# Structure of the stress-related LHCSR1 complex determined by an integrated computational strategy

Ingrid Guarnetti Prandi[1,2,3], Vladislav Sláma[1,3], Cristina Pecorilla [1], Lorenzo Cupellini [1✉] & Benedetta Mennucci [1✉]

Light-harvesting complexes (LHCs) are pigment-protein complexes whose main function is to capture sunlight and transfer the energy to reaction centers of photosystems. In response to varying light conditions, LH complexes also play photoregulation and photoprotection roles. In algae and mosses, a sub-family of LHCs, light-harvesting complex stress-related (LHCSR), is responsible for photoprotective quenching. Despite their functional and evolutionary importance, no direct structural information on LHCSRs is available that can explain their unique properties. In this work, we propose a structural model of LHCSR1 from the moss *P. patens*, obtained through an integrated computational strategy that combines homology modeling, molecular dynamics, and multiscale quantum chemical calculations. The model is validated by reproducing the spectral properties of LHCSR1. Our model reveals the structural specificity of LHCSR1, as compared with the CP29 LH complex, and poses the basis for understanding photoprotective quenching in mosses.

[1] Department of Chemistry and Industrial Chemistry, University of Pisa, 56124 Pisa, Italy. [2] Present address: Department for Innovation in Biological, Agro-Food and Forest Systems, DIBAF, University of Tuscia, Via S. Camillo de Lellis s.n.c., 01100 Viterbo, Italy. [3] These authors contributed equally: Ingrid Guarnetti Prandi and Vladislav Sláma. ✉email: lorenzo.cupellini@unipi.it; benedetta.mennucci@unipi.it

In plants and algae, sunlight is collected by light harvesting complexes (LHCs) and rapidly funneled to the reaction centers of photosystems. In response to varying light conditions, LHCs can also play photoregulation and photoprotection roles[1,2]. By sensing directly or indirectly the excess energy in the thylakoid membrane, these complexes can switch to a photoprotective state where the excitation energy is dissipated into heat, in a process called nonphotochemical quenching (NPQ)[3,4]. Both light-harvesting and quenching processes are determined by the embedded pigments, chlorophylls (Chls) and carotenoids (Cars), and their interactions.

While the details of NPQ are still not completely understood, it is now well established that the trigger for the LH-to-quenching switch is a change in the pH of the lumen side of the thylakoid membrane embedding the photosystem. However, the way the pH change is felt and used strongly depends on the type of photosynthetic organism. In particular, algae express the so-called light-harvesting complex stress-related (LHCSR) proteins[5], which are responsible for quenching the excess energy as part of the NPQ mechanism.[6] It seems that LHCSR proteins can directly sense pH changes in the thylakoid lumen and activate the quenching process at low pH[7,8]. A different strategy is used in plants, where a pigment-less protein (called PsbS) is responsible for sensing the lumenal pH and activating the quenching through an interaction with the LHCs. Mosses, instead, have both PsbS and LHCSR proteins active in NPQ[5,9–12]. This is a very interesting aspect as mosses represent evolutionary intermediates between algae and plants.

While high-resolution X-ray (or Cryo-EM) structures have been solved for most LHCs, no structural information is available about the LHCSR family. Knowing the structural details of the protein and of the embedded pigments is fundamental to understanding the functional characteristics of LHCSRs. As these proteins are central in the evolution of photoprotection strategies passing from algae to plants, clarifying their structure is extremely important to understand how photoprotection evolved in photosynthetic organisms when passing from water to land.

Here we employ a modeling strategy to build, refine, and validate the structure of LHCSR1 of the moss *Physcomitrella patens* (Pp) for which not only the structure but also the exact pigment composition is still unknown. Despite the assumed structural similarities with the LHCs of Photosystem II, the properties of LHCSR1 are markedly different as also shown through single-molecule spectroscopy[13,14]. Here, to uncover these differences, we combine homology modeling and extensive molecular dynamics refinement, including enhanced sampling techniques. The proposed structure is thoroughly validated by means of multiscale quantum mechanical calculations of its optical and chiro-optical spectra and finally compared with the closest known LHC structure, the CP29 complex. Our strategy reveals similarities and unexpected differences between LHCSR1 and CP29: while most of the conserved regions share high structural similarity to CP29, LHCSR1 deviates in the less conserved lumenal side. We highlight important differences in the mutual orientation of a strongly coupled Chl cluster, that reflect in the spectral properties of LHCSR1.

## Results and discussion
### Sequence alignment and initial LHCSR structure.
Comparison of Pp-LHCSR1 with other LHCs reveals that the most similar amino acid sequences with an available crystal structure are LHCII[15] and CP29[16], with 32.2 and 31.2% sequence identity, respectively. Although LHCII has a slightly higher amino acid sequence similarity, CP29 was chosen as a template due to the loop length in the lumenal region, which matches LHCSR1.

Aided by the conserved binding motifs associated with the transmembrane helices and chlorophyll-binding sites, we obtained the consensus sequence alignment between LHCSR1 and CP29 shown in Fig. 1a. Together with the main Chl binding motifs, we note that the charged residues involved in salt bridges among transmembrane helices are conserved. The sequences of LHCSR1 and CP29 share high identity in the last part of the stromal loop preceding helix A, and more importantly, with helices A and B which are the core of the protein and bind most of the pigments. On the other hand, a lower identity is found for helix C, and for the small amphiphilic helices D and E.

Surprisingly, we found that the Neo-binding Tyr in the lumenal loop of CP29 is also conserved in LHCSR1. This binding site, namely N1, was excluded in other homology modeling studies of LHCSR[17], because a different alignment was predicted. However, this Tyr is part of a highly conserved motif among LHCSR1 and LHCSR3 (Supplementary Fig. 1), which leads us to believe that this amino acid position/orientation is indeed conserved between LHCSR and CP29.

Experimental studies have suggested that LHCSR1 binds only Chl *a*[13,18–20], in contrast to LHCSR3 which has a 7/1 chlorophyll a/b ratio[17,21]. For this reason, we considered all Chls as Chl *a* in our model. Based on the conserved Mg binding sites and previous literature[14,17–19], Chls 602, 603, 609, 610, and 612 should be present on the stromal side of the complex, and Chls 604 and 613 should be bound to the lumenal side. Also, Chl *a*611 is thought to be present[13,14,22], even if it is not directly bound to a protein residue. For this reason, we retained also Chl *a*611 in the LHCSR model, along with its axial ligand, the DPPG lipid[16].

When considering the carotenoid content of LHCSR, some divergences are present in the literature. Sites L1 and L2 are assumed to be occupied by a xanthophyll as in all LHC proteins. However, the identity of the bound carotenoids varies between different LHCs[1]. It was suggested that site L2 is occupied by Vio, and site L1 is occupied by Lut, in both LHCSR1 and LHCSR3[17,20], as occurs in CP29[16].

There are two "external" carotenoid binding sites in LHCII, N1, and V1[15]. The absence of Neo in LHCSR1 led us to consider Lut as a possible candidate for the N1 site, as suggested by Pinnola et al.[20], and by the possible presence of Lut in the N1 site of LHCI[23]. Pinnola et al. also suggested that four carotenoids are present per LHCSR1 apoprotein[20]. However, we exclude that the V1 site can be occupied in LHCSR because it is only stable in trimeric LHCII complexes[24]. We, therefore, included a Lut in the N1 site in addition to the aforementioned L1 Lut and L2 Vio.

The final model is shown in Fig. 1b superimposed with CP29. The predicted secondary structure is virtually identical to CP29, except for the longer loop which protrudes in the stromal side, and for the junction between helices A and D. Notably, Tyr136, which in our alignment corresponds to the N1-site Tyrosine binding Neo in CP29, Tyr135[16]. In our model, this Tyr is predicted in the same position as in CP29, and it is close to the N1 lutein.

### Refinement of the structure through molecular dynamics.
Although homology modeling can be a good starting point for protein structure prediction, it is clear that differences in the amino acid sequences between the template and the built model, especially in helices D and E, can affect the structure of the complex and should be further refined. Therefore, to obtain a more accurate and dynamic model, we employed a molecular dynamics (MD) protocol.

Six 1 μs-long trajectories (MD1–MD6) were simulated starting from the homology model embedded in a lipid membrane (see "Methods"). Given the higher homology in the transmembrane helices than in the stromal and lumenal parts, we expect that the

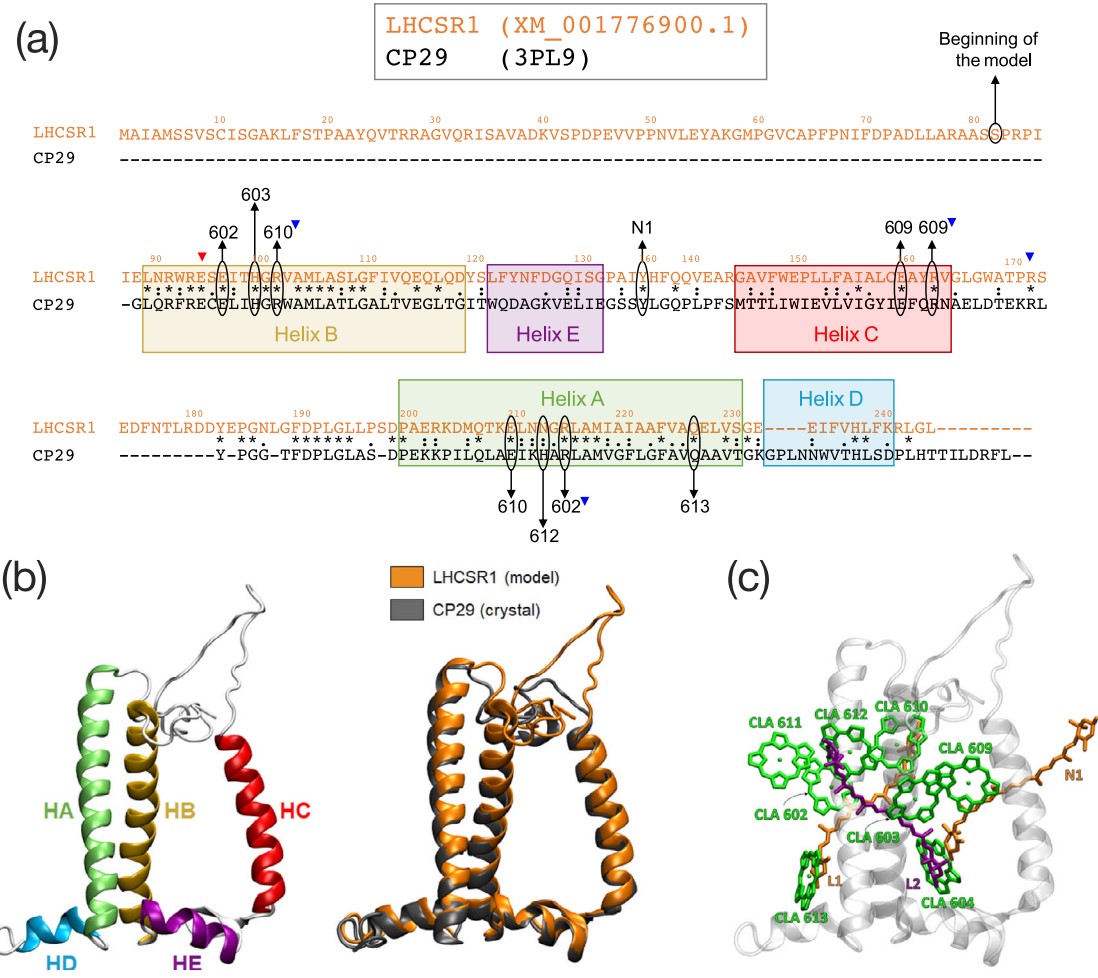

**Fig. 1 Structural details of LHCSR1 after homology-modeling. a** Sequence alignment between CP29 (*Spinacia oleracea*) and LHCSR1 (*Physcomitrella patens*). The color squares project the CP29 α-helices in the LHCSR1 sequence after the homology modeling. "*" indicates conserved residues, ":" indicates strong similarity, while "." indicates weak similarity. Conserved binding sites of the Chls are indicated with the Chl number, with a downwards-pointing blue triangle indicating the salt-bridge Arg residue participating in the binding site. The N1 binding site of CP29 is also indicated with "N1". Additional conserved residues participating in salt bridges on the stromal side are indicated with downwards-pointing triangles (red/blue for acidic/basic residues). **b** Helices nomenclature and comparison of the model of LHCSR1 as predicted by homology modeling (orange) with the parent structure CP29 (PDB: 3PL9)[16]. **c** Representation of the pigment arrangement in the modeled LHCSR1 system: chlorophylls (all of type "a" - CLA) are represented in green, luteins in orange, and violaxanthin in purple.

model is more accurate in the transmembrane region. For this reason, the system was pre-equilibrated for 100 ns with restraints on the transmembrane domains, while letting the stromal and lumenal regions equilibrate freely without perturbing the core of the protein.

The six production trajectories, propagated without restraints, show stable transmembrane helices. However, they present significant deviations of the lumenal region and especially unfolding and refolding of helices D and E (Supplementary Figs. 2 and 3). This result led us to follow a further equilibration protocol employing accelerated MD (aMD). Also, in this case, weak restraints were added to the parts of the protein for which homology modeling gave greater confidence. The aMD equilibration was started from the end of MD1, and the coordinates of the complex after 400 ns were extracted and used as a starting point for two further replicas (MD7–MD8). Specifically, the extracted structure underwent the same protocol as the initial homology modeling structure.

The ensemble of structures obtained in the MD refinement was characterized by analyzing the Ramachandran angles of the apoprotein with a hierarchical clustering algorithm (see "Methods").

By analyzing the RMSD in the Ramachandran space between frames of all trajectories (Supplementary Fig. 4), we note that frames within each trajectory are more similar to each other than with different replicas. However, replicas MD7 and MD8, obtained after aMD equilibration, are much closer to each other, suggesting that the aMD equilibration step has allowed the system to relax towards a more stable state.

For the clustering, we used Ramachandran dihedral angles of the helical regions only, because the mobility of the loops results in a high level of noise. Only the last 500 ns of each trajectory were included in the clustering. We included all MD trajectories in the clustering, which is "blind" to the equilibration protocol used. The structures are clustered together only based on the Ramachandran angle similarity.

The four obtained clusters are shown in Fig. 2a. The clustering algorithm has distributed the frames from trajectories MD1-MD6 in three different clusters (Clusters 1–3) while the structures obtained from MD7 and MD8 have been merged in the same cluster (Cluster 4) thus indicating a clear separation of these replicas, which have been pre-equilibrated with aMD, from the others. Notably, each replica MD1–MD6 contributes to only one of clusters 1–3, meaning

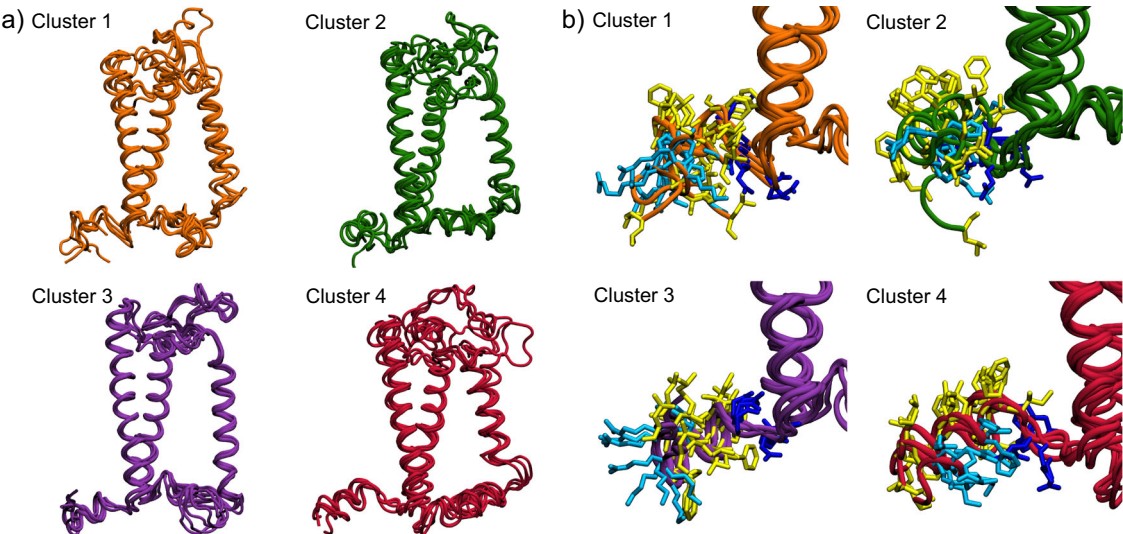

**Fig. 2 Molecular dynamics structures of LHCSR1 obtained from clustering of the backbone conformations. a** Entire protein structure. The clusters differ mainly in the folding and orientations of helix D (HD) and helix E (HE). **b** Focus on Helix D. The apolar amino acid residues are represented in yellow, the charged residues in blue, and the polar ones in cyan.

that our conventional MD replicas tend to diverge from the initial state and get stuck in different metastable states.

The clusters mainly differ in the folding of helices D and E, and partially in the lumenal part of the transmembrane helix C. The orientation of helix D also widely differs among the clusters. In clusters 1 and 3, helix E is partially or completely unfolded, suggesting that it has not yet reached a completely stable equilibrium. Moreover, in clusters 1 and 2 helix D is partially unfolded. The large variance observed for helices D and E suggests that these regions of the protein, as predicted by homology modeling, have not reached the free energy minimum. This is not surprising, considering the low homology between LHCSR1 and CP29 for these helices. On the other hand, the structures of Cluster 4 show a more stable picture for the lumenal side. In Cluster 4, in fact, helix E is mostly folded and helix D is in a different orientation than the other clusters (see below). Cluster 4 is also characterized by a shortened helix C, which is partially unfolded in the lumenal side.

Experimental studies performed on LHCSR proteins from other species[17] indicate that the D helix acts as a pH sensor in the lumen and can respond by tuning NPQ according to the acidity of the lumenal layer. Helix D is amphiphilic and should be oriented with the more polar residues toward the exterior of the membrane. Comparing the clusters (Fig. 2b), it is clear that the conformation of helix D is closer to our expectation only in Cluster 4. Figure 2 shows that in the structures of this cluster the apolar amino acid residues (represented in yellow) point towards the membrane, and the charged (blue) or polar (cyan) residues are oriented towards the lumen interface, allowing the interaction with water.

Only by accurately equilibrating this helix is it possible to assess the position of acidic residues and their possible role in the regulation of quenching. As shown in Fig. 2b, in Cluster 4 the acidic residues are found at the turn connecting helices A and D. Protonation of these residues may result in a conformational change involving these two helices that might promote quenching in the close L1 site.

We expect that the variations found on the lumenal side of the protein have an effect on the pigment positions and orientations. As a matter of fact, Chls a604 and a613 showed larger variations, as they are close to helices D and E respectively. Instead, the

chlorophylls on the stromal side present only small variations from the initial model and among the different clusters (Supplementary Figs. 5–9). The interactions between the Chls and their axial binding sites revealed stable binding of the Chls (Supplementary Fig. 10). We note that in MD7 there are some differences in the distance between Chl a613 and its binding site, the Gln226 residue (Supplementary Fig. 10), which is due to the presence of a water molecule between the residue and the chlorophyll, causing chlorophyll a613 to be indirectly bound to Gln in some time windows (Supplementary Fig. 11). This conformation is further stabilized by a hydrogen bond between the O1D atom of the chlorophyll and Ser 230. Also, Chl a604, which is not axially bound to protein residues, remained close to the helix B backbone in all simulations except MD1 (Supplementary Fig. 12). Analogously to CP29 and LHCII[25], a604 is bound to the backbone oxygen of Gly109 through a water molecule. Finally, the axial binding between a611 and the DPPG lipid remained stable and tight in all simulations except MD1 and MD3 (Supplementary Fig. 12).

Distributions of mutual distances and orientations between the Chls were generally similar to CP29 (Figs. S5–S9). The largest difference was found for the mutual orientation of Chls a603–a609 (Supplementary Fig. 6), for which the more thoroughly equilibrated MD7 and MD8 (Cluster 4) showed a more linear orientation of the dipoles with respect to CP29 and to the other MDs. Also for the a602–a603 pair (Supplementary Fig. 8), we notice that MD7 and MD8 deviate more from CP29, suggesting that Chl a603 has a different orientation in LHCSR, which is captured only after aMD equilibration. Figure 3 summarizes the comparison between cluster 4 LHCSR1 and CP29 for the most coupled Chl pairs. Contrary to the cluster a602-a603-a609, the mutual orientation within the strongly interacting Chl a611–a612 pair is only slightly altered from CP29 (Fig. 3).

Carotenoids in internal sites L1 and L2 were stable throughout the simulations and we have not observed significant displacements, in agreement with previous MD simulations on LHCII and CP29[26–28]. The only external carotenoid is lutein in the N1 site, as suggested by our alignment (see above). N1-Lut remained in a stable binding site in the majority of simulations, except for two replicas, MD4 and MD6 (Supplementary Fig. 13). In MD6 the lutein interacts with the stromal side of the complex

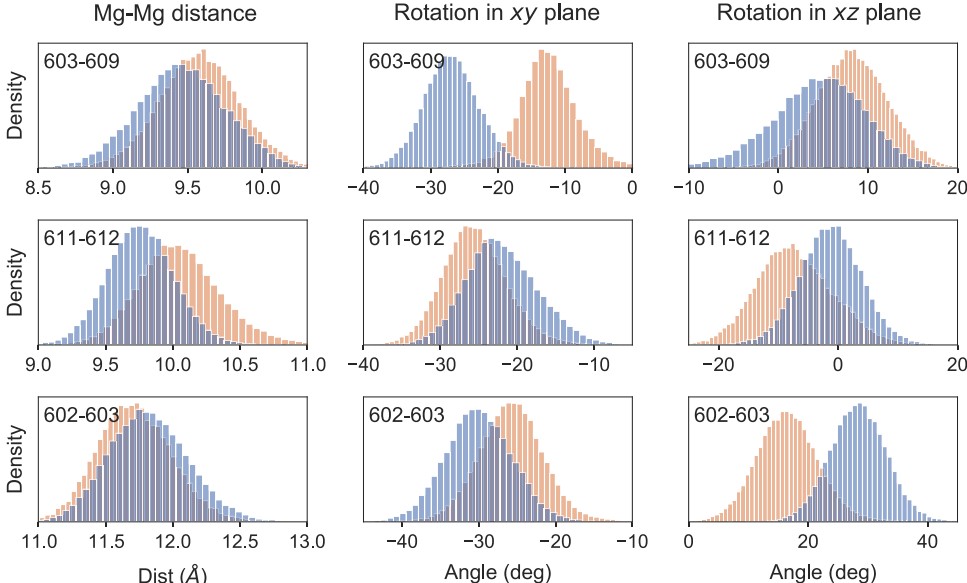

**Fig. 3 Comparison of mutual distances and orientations for the chlorophyll strongest coupled pairs in LHCSR1.** LHCSR1 Cluster 4 is shown in orange and CP29 is shown in blue. The first column corresponds to the Mg–Mg distance between chlorophylls, the second column to the mutual rotation of the chlorophylls in the plane defined by the chlorophyll ring, and the last column corresponds to a rotation in the plane perpendicular to the chlorophyll ring. The analysis was performed on 40,000 configurations extracted every 50 ps of the MD simulation for LHCSR1 and on 30,000 configurations with the same time-step for CP29.

and is found close to Chls a603 and a609. In MD4, instead, the N1-Lut moves towards the stromal side of the complex, without interacting with the Chl-binding sites. The unbinding of N1-Lut is only observed in those MDs where the lumenal loop is not completely equilibrated. Conversely, in the trajectories started from aMD equilibration, N1-Lut is stably bound. Therefore, the loss of lutein from the N1 site could also be ascribed to an incomplete equilibration of the lumenal side of the complex. Experimentally, the external carotenoid binding sites of LHCSR are still unclear and we cannot exclude that the binding mode of this carotenoid in LHCSR is slightly different.

When we consider the mutual orientation of the carotenoids and Chls, the largest differences between LHCSR1 and CP29 were observed for Chls a604 in N1 and a613 in L1. These Chls are far from the other Chl clusters and close to the helix D and E. Therefore the orientation of the Chls is dependent on the equilibration and folding of the helices. On the other hand, the mutual orientation of the L1-Lut with Chl a612 and L2-Vio with Chl 603, suggested as possible quenching sites in the CP29, is very similar between the LHCSR1 and CP29 (Fig. 4). The distribution of the mutual angles is only slightly broader for LHCSR1 due to the missing pigments compared to CP29, which leads to larger freedom in the mutual orientation.

**Validation of the structure through spectroscopy simulations.** In order to validate our LHCSR1 model, we computed absorption and circular dichroism (CD) spectra on structures extracted from the eight MD replicas and grouped them according to the previous clustering algorithm. We excluded from these calculations all structures coming from MD6, due to the fortuitous interaction of N1-Lut with the a603–a609 dimer, which we regard as unphysical and irrelevant for LHCSR1. All the spectra (shown in Supplementary Fig. 14) were obtained using an excitonic model (see Methods).

As it can be seen from the plots, the absorption spectra of the four clusters are very similar to each other, as the majority of pigment–protein and pigment–pigment interactions are conserved in different protein conformations. However, these results

also suggest that absorption spectra are not very sensitive to the structure, and we cannot use them to effectively discriminate between clusters. CD spectra, on the other hand, present more variability among clusters, especially in the position of the main negative band and in the intensity of the weak positive band. Even though these differences are not extreme, they can be used to distinguish the various clusters.

The differences observed between the clusters' spectra could be explained by differences in the respective excitonic properties. Excitonic couplings were essentially the same in all clusters (Supplementary Fig. 15), the only difference being a slightly larger a603–a609 coupling in Cluster 4. These differences, however, are too small to explain the effects on the positions and intensities of the CD band. We instead found a much larger variability of the site energies (Supplementary Fig. 16a). The largest variability between the clusters was found for Chls a604 and a610. The site energy of Chl a610 is influenced by the vicinity of the highly mobile stromal loop (Supplementary Fig. 17) Therefore, for Chl a610 the largest differences between the clusters are due to the environmental contribution to the excitation energy. On the other hand, the site energy of Chl a604 is influenced by a hydrogen bond with the lumenal loop backbone nitrogen (Supplementary Fig. 18), which leads to comparable geometrical and environmental contributions to the site excitation energy.

Comparison with experiments shows that Cluster 4 best reproduces the optical spectra. This is expected, because Cluster 4 includes the replicas that were better equilibrated by means of accelerated MD, and showed better overall folding of helices D and E. The experimental CD shape is well reproduced in Cluster 4, especially in the position of the main negative band, and in the shape of the weak shoulder at ~14,400 cm$^{-1}$. Due to this better behavior, the following detailed analysis of LHCSR1 spectra and the comparison with CP29 will be based on Cluster 4 only.

Let us start the analysis with the exciton structure of LHCSR and compare it to the one of CP29. The exciton Hamiltonian of LHCSR1 and CP29 was computed as described in the Methods section; for CP29 the structures extracted from the MD presented in ref. [28] were used.

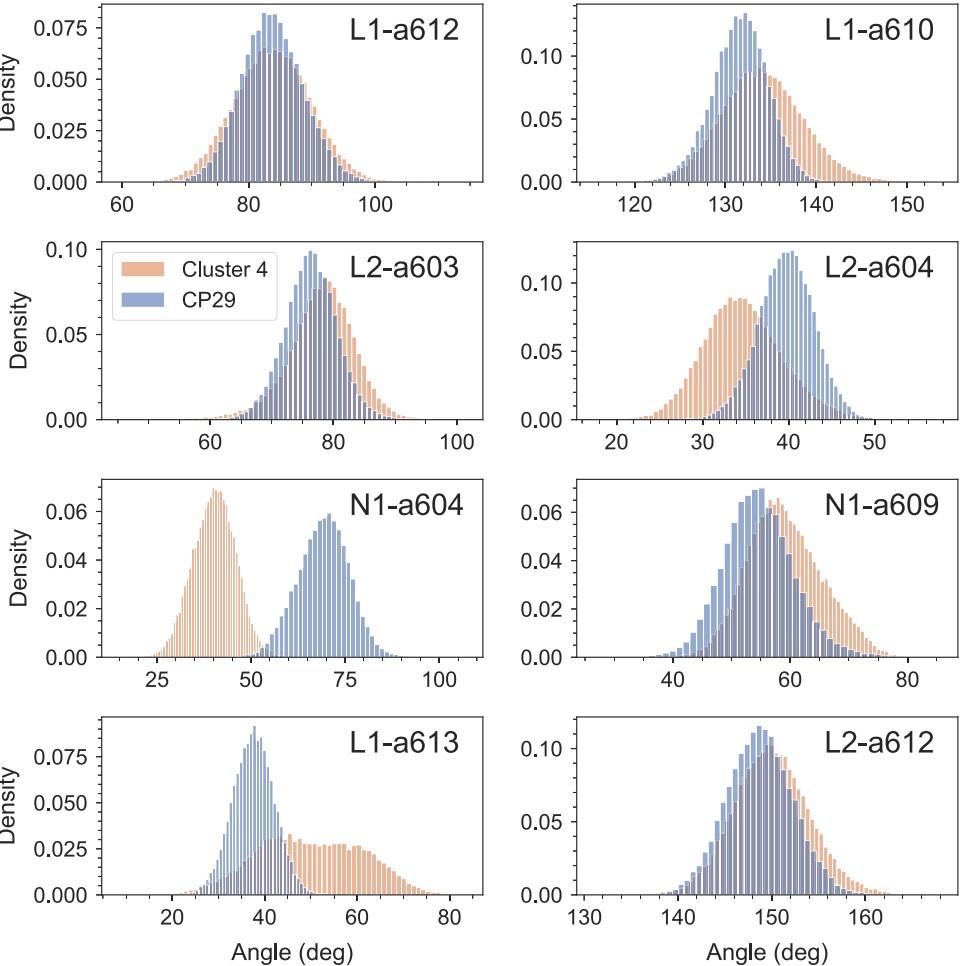

**Fig. 4 Mutual orientation of Cars and Chls.** The orientation is defined as the angle between Chl NB → ND vector and Car C15 → C6 vector corresponding to the orientation of the Chl $Q_y$ transition and Car $S_2$ transition, respectively. The analysis was performed on 40,000 configurations extracted every 50 ps of the MD simulation for LHCSR1 (orange distribution) and on 30,000 configurations with the same time-step for CP29 (blue).

The results, reported in Fig. 5, reveal for both complexes the formation of two main clusters of strongly interacting Chls, a602-a603-a609, and a610-a611-a612, which contribute to both the lowest and the highest exciton states. The lowest exciton states of the CP29 are slightly blue-shifted with respect to the LHCSR1. Another interesting finding for LHCSR1 is the role of the Chl a604. This Chl has the lowest site energy, but because it is separated from the other Chls it retains its energy and therefore it is above the lowest exciton states formed by the strongly interacting clusters. A similar situation is true for Chl a613. Both these Chls are in between the high energy exciton states and the exciton states with the lowest energy, therefore they might play a role as a bottleneck in the energy transfer dynamics of this system[29].

We now compare the simulated optical spectra of LHCSR1 and CP29 (Fig. 6). The simulated absorption spectra show a very good agreement with the experimental ones, including the small red-shift of the LHCSR1 absorption spectrum with respect to CP29. This red-shift can be explained by different site energies and the consequent shifts of the exciton states (Fig. 5). The major contribution to the CD spectra of both systems comes from the two strongest interacting Chl pairs a611–a612 and a603–a609.

The blue-shift of the LHCSR1 CD spectrum, opposite to the absorption spectra, is also reproduced by our simulations. It can be partially explained by slightly different mutual orientations of the transition dipoles (Fig. 3). Notably, the a603–a609 pair in LHCSR1 differs substantially from CP29 only in Cluster 4, further

indicating that the equilibration with aMD allows reaching more realistic structures.

Another contribution to the difference in the CD spectra of CP29 and LHCSR1 is due to the blue-shift of the Chl a611 site energy in CP29 (Supplementary Fig. 16). Such a blue-shift is caused by a positively charged Lys199 in the vicinity of the Chl a611 in CP29, whereas LHCSR1 features an Asn211 in that position. The shift of the Chl a611 site energy reduces the exciton mixing with Chl a612 and lowers its contribution to the CD spectra of CP29.

**Investigating alternative pigment compositions.** From the analysis reported in the previous section, it clearly appears that our simulated structure of LHCSR1 correctly reproduces the measured spectra including the small differences with respect to CP29. This good agreement can be further exploited to investigate the pigment composition of LHCSR1 which in the literature still represents an open question.

Most of the chlorophylls we have considered in our LHCSR1 model are bound to a conserved amino acid of the protein, except for a611 and a604, whose presence in LHCSR1 cannot be deduced from the conserved amino acids[22]. Contrary to a604, Chl a611 is excitonically coupled to Chl a612, and this coupling is thought to be the origin of the CD spectral shape in all LHCs[17,30]. We, therefore, investigated the effect of the Chl a611–a612 interaction on the spectra by removing it from the exciton system (Fig. 7a, b). The resulting absorption spectrum is broader and has a lower

intensity due to the different redistribution of the transition dipoles and exciton energies. More substantial effects can be seen on the CD spectra, where the main negative peak is considerably weaker and broader. We estimate that the Chl pair a611–a612 is responsible for ~40% of the negative peak intensity in the CD spectra.

Another open issue is the presence of Chl b in some binding sites of LHCSR1. Pigment analysis suggests that LHCSR1 does not bind Chl b[18–20], but a recent analysis in terms of spectral deconvolution seems to suggest the presence of one or more Chls

b[22]. Based on the LHCSR3 pigment analysis[17] and on the presence of Chl b609 in LHCII[15] we simulated the substitution of Chl a609 with a Chl b in LHCSR1 (Fig. 7c, d). The Chl-a from the original model was replaced by Chl-b for the simulation of the optical spectra by blue-shifting the original Chl-a site energy by +440 cm$^{-1}$ and reducing the transition dipole and corresponding couplings by a factor of 0.78. Both the energy shift and the scaling factors were obtained by comparing the spectral properties of Chl a and Chl b in solution[31,32].

The simulated absorption spectrum with b609 showed a blue-shift and reduction of the intensity compared to the full Chl a system. The CD spectrum was blue-shifted as well, and the intensity of the main negative peak was lowered by 20% compared to the full system (Fig. 7d). The effect on the spectrum is due to a reduction of the exciton delocalization between Chls 603 and 609 when Chl a609 is substituted by a Chl b, which has a significantly higher excitation energy. Our calculation is based on the assumption that Chl b609 would have the same orientation as the corresponding Chl a609. However, the main effect of reducing exciton delocalization does not depend on the exact position or orientation of Chl b. From this analysis, we can conclude that the simulated spectra for the all-Chl a system agree with the experimental spectra much better than the ones simulated for the system with Chl b609.

Another open issue is the number and the type of carotenoids in LHCSR1. As a first test, we have artificially removed all the carotenoids from the exciton Hamiltonian and recalculated the spectra. The results are reported in Fig. 7g for absorption and Fig. 7h for CD. From these figures, it is evident that the carotenoids have a significant effect on the CD spectrum. In particular, they are responsible for the non-conservative character of the spectrum in the Chl-a region[33] and for the intense negative peak through off-resonant exciton interactions. We took advantage of this fact to assess the position of lutein in the N1 site. Removing the N1-Lut from the exciton system leads to a lower intensity of the main negative peak (Fig. 7e, f), suggesting that this lutein does contribute to the CD spectrum. This analysis shows that a carotenoid is present at the position predicted by our model, or at a similar one.

## Conclusions

We have used an integrated computational modeling strategy to predict and validate the structure of LHCSR1 in its natural environment. The protocol is made of three subsequent steps: construction of a first guess from homology modeling, refinement of the guess with molecular dynamics and clustering, and final

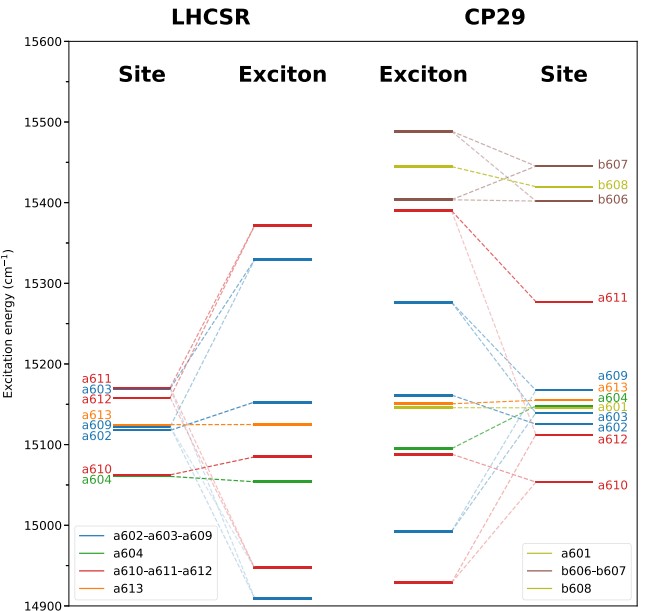

**Fig. 5 Exciton structure of LHCSR1 and CP29.** Excitation energies of the individual pigments (site) are compared with the exciton states of the whole system. The dashed lines represent the contribution of the site basis to the exciton states and their opacity the magnitude of the contribution. Different colors of the lines represent individual clusters of the strongly interacting chlorophylls. The excited states of LHCSR1 were compared with the ones from CP29. The exciton states were computed from the average Hamiltonian of cluster 4 and single replica for the current model of the LHCSR1 and CP29 respectively. The averaged Hamiltonians and exciton energies for both systems are included in Supplementary Tables 1–3.

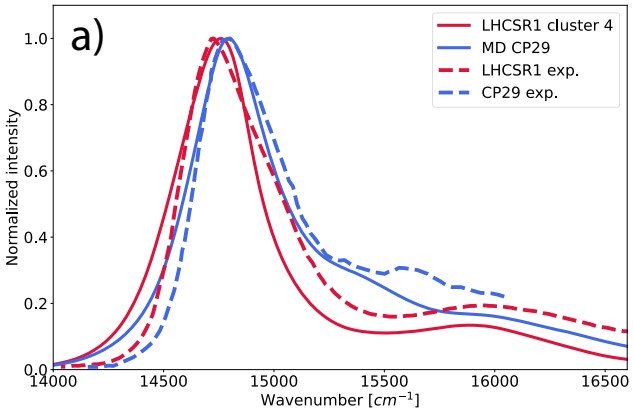
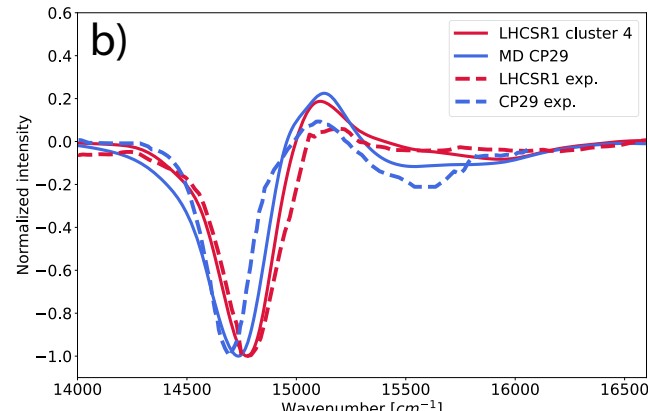

**Fig. 6 Comparison between LHCSR1 and CP29 optical spectra.** Spectra computed for the LHCSR1 cluster 4 and CP29 (solid lines), compared with the experiment (dashed lines). **a** absorption spectra, **b** CD spectra. Note that calculated Chl site energies were all shifted by the same amount (−1381 cm$^{-1}$) to account for the error of the quantum chemical method; this shift was determined independently on LHCII (see "Methods"). The optical spectra for the cluster 4 of the LHCSR1 were obtained by averaging over spectra of 102 configurations (replicas MD7 and MD8) and for CP29 by averaging over 79 configurations. The experimental spectrum for LHCSR1 was taken from ref. [20] and the CP29 one was taken from ref. [52].

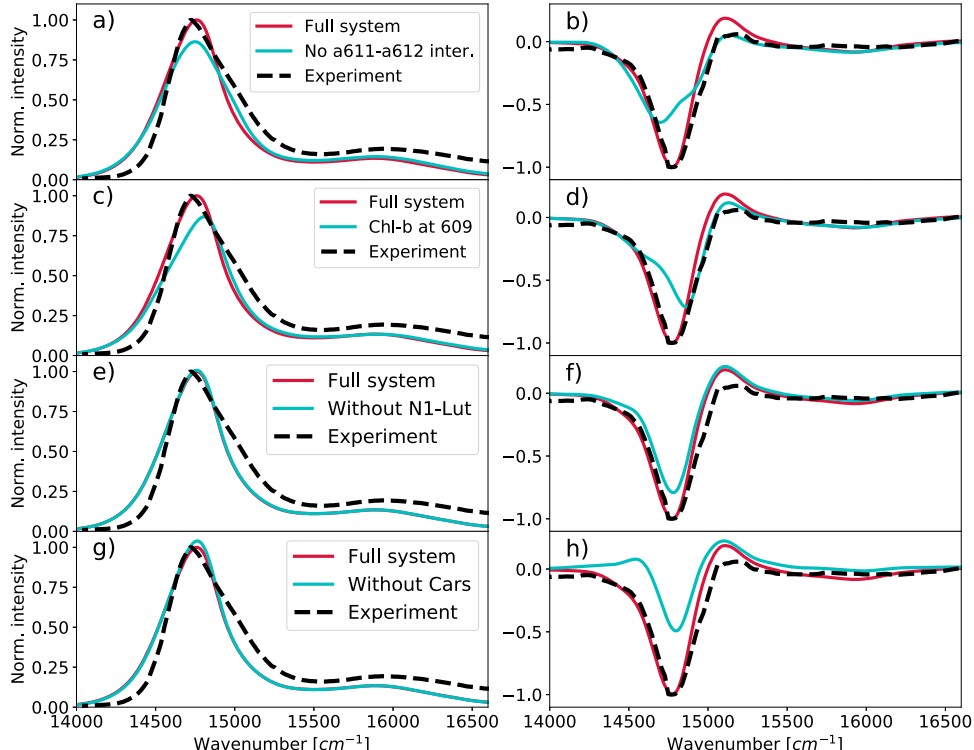

**Fig. 7 Effect of individual pigments on the optical spectra of LHCSR1. a** Absorption and **b** CD spectra for the LHCSR1 where the interaction between Chl a611-a612 was set to zero. **c** Absorption and **d** CD spectra for the LHCSR1 where the Chl a609 was replaced by the Chl-b. **e** Absorption and **f** CD spectra of the LHCSR1 without N1-Lut. **g** Absorption and **h** CD spectra of the LHCSR1 without all the carotenoids. The optical spectra were obtained by averaging over spectra of 102 configurations (replicas MD7 and MD8). The experimental spectra (black dashed lines) were taken from ref. [20].

validation with quantum chemical simulations of absorption and CD spectra.

We have seen that the MD refinement is fundamental, as it allows the system to deviate from the model used for the homology modeling (here CP29), especially in those regions that present a lower sequence homology, here the small amphiphilic helices D and E. The folding and orientation of these helices cannot be easily predicted from homology, also because their orientation is influenced by the interface between membrane and solvent. We have also seen that enhanced sampling techniques, such as the accelerated MD employed here, can be necessary to obtain fully equilibrated structures. Finally, spectral modeling with multiscale quantum chemical tools has shown to be a crucial step in this strategy, as it allows validating the model in a completely independent way. Indeed, our simulated structures not only well reproduce the shape of absorption and CD spectra, but also correctly predict the spectral changes from the parent CP29.

Although here we focused on LHCSR1 of *Physcomitrella patens*, the high homology among LHCSR proteins suggests that our results also apply to LHCSR1 and LHCSR3 of *Chlamydomonas reihnardtii*. Perozeni et al. have presented homology models of LHCSR1 and LHCSR3 from *Chlamydomonas rein-hardtii* based on Lhcb1 as a template[22]. As in our case, their homology models follow very closely the template structure. Our MD results however suggest caution in using these models, as the real structure of LHCSR may deviate substantially from the currently available template structures.

More in general, the protocol proposed here promises to be a viable strategy for the modeling of LHC proteins for which a high-resolution structure is not known experimentally. As LHCSRs represent a cornerstone in the evolution of the photo-protective function, we believe that our structure will be the basis for new investigations into the evolution of LHC antennae.

## Methods

**Homology modeling**. Primary sequence alignment was first performed separately for Lhcb proteins (LHCII, CP29, and CP26) and LHCSR proteins (LHCSR1 of *Physcomitrella patens*, Pp, LHCSR1, and LHCSR3 of *Chlamydomonas reihnardtii*) in order to pinpoint the most conserved regions, which contain conserved motifs belonging to the three transmembrane helices and to pigment binding sites. Sequence alignment was performed with ClustalX[34]. The sequence of Pp-LHCSR1 (coding sequence: XM_001776900.1) was aligned with LHCII and CP29 in order to find the best match. The CP29 alignment was chosen as a model on the basis of a shorter length of the lumenal loop, more similar to LHCSR. The final alignment between LHCSR1 and CP29 was performed in two steps: first, LHCSR was aligned to CP29 from Gly88 up to Leu172, comprising in this way helices B, E, and C. Then it was aligned from Tyr173 to Thr237, comprising the stromal loop and helices A and D. These two alignments were combined and used in order to ensure that all conserved binding sites were aligned[35]. Structural modeling was performed with Modeller using the final alignment to CP29, along with the CP29 structure from spinach (PDB ID: 3PL9). Cofactors were chosen taking into account key binding pockets and/or amino acids that are related to the stability of similar light-harvesting antenna systems, also considering experimental data about chlorophyll/carotenoid ratio in LHCSR complexes. The conserved cofactors were explicitly included in the homology modeling, in order to avoid physically unsound side-chain conformations. Eight chlorophylls *a* were kept from CP29, i.e., 602, 603, 604, 609, 610, 611, 612, and 613, based on previous literature. Lutein and violaxanthin were kept in sites L1 and L2 respectively (see Fig. 1). For convenience, in this work, chlorophylls, carotenoids, and the protein helices are named following CP29/LHCII crystal structures[15,16,25] (see Fig. 1b). The first amino acids from the CP29 N-terminus loop share a very low homology with LHCSR1 and cannot be used as a template for homology modeling. For this reason, the N-terminus was discarded after building the LHCSR1 model. Our model thus starts at Ser82, based on the original sequence numbering (see Fig. 1a). After building the LHCSR model, a lutein was added to the N1 site. The coordinates of the N1-bound neoxanthin ring were taken from CP29, and the rest of the lutein was built based on templates. The coordinates of the DPPG lipid which ligates Chl a611 were taken from CP29. For adding these pigments, we used the CP29 coordinates from the Cryo-EM structure of PSII (PDB: 3JCU)[36].

**Molecular dynamics**. The *tleap* module of AmberTools was employed to add hydrogens to prepare the LHCSR structure. All titratable residues were kept in their standard protonation state, except for Chl-binding histidines, which were δ-pro-tonated in order to allow Mg binding. This protonation was chosen on the basis of

pKA calculations with the H++ server[37] at pH 7. After a first *in vacuo* mini-mization with a 10 kcal mol$^{-1}$ Å$^{-2}$ harmonic restraint on backbone atoms, LHCSR was embedded in a DOPC bilayer membrane following the same alignment achieved in a previous MD simulation by some of us and solvated with water molecules. For all MD steps, the AMBER ff14SB force field was used for the protein. Carotenoids were described with the force field by Prandi et al.[38], and chlorophyll *a* was described with the force field by Ceccarelli et al.[39] with Zhang et al. modifications[40]. Lipids were described with the lipid14 force-field[41], and water was described with the TIP3P model. The MD simulation protocol was based on a previous study on CP29[28]. A first minimization was performed only on the lipids that made close contact with the protein or cofactors. Then, the entire system was minimized without constraints. A 5 ps simulation in the NVT ensemble fol-lowed by a 100 ps simulation in the NPT ensemble were used to heat the system to 300 K, with the protein and cofactors constrained by a 10 kcal mol$^{-1}$ Å$^{-2}$ har-monic restraint. The box equilibration step was performed in the NPT ensemble by gradually releasing the restraints to 0.4 kcal mol$^{-1}$ Å$^{-2}$ in 5 ns. An additional 100 ns simulation was performed to equilibrate the loops and other mobile regions of the protein, restraining only the backbone of the transmembrane helices by a 0.4 kcal mol$^{-1}$ Å$^{-2}$ harmonic restraint. The production simulations (replicas MD1–MD6) were performed freely for 1 µs in the anisotropic NPT ensemble. The Langevin thermostat and (for NPT simulations) the Monte Carlo anisotropic barostat were used to control temperature and pressure, respectively. The SHAKE algorithm was used in all simulations along with a 2 fs time step. Particle-mesh Ewald electrostatics with a 10 Å non-bonded cutoff was used.

### Accelerated MD equilibration.
Starting from the final structure of the MD1 replica, we performed a second equilibration using accelerated MD (aMD)[42,43]. aMD is an unconstrained enhanced sampling method that raises the potential energy of the system around minima, making the crossing of free-energy barriers more likely. We use aMD to achieve a further equilibration of the amphiphilic and loop regions of the protein, adding distance restraints in the AMBER style to the cofactor binding sites and among the transmembrane helices in order to keep their secondary structure. Dual-boost aMD simulations were run, biasing the dihedral angles and on the total energy. The parameters of the bias potential were derived as suggested for membrane proteins[44]. Three replicas of 1 µs each were simulated in those conditions, and one structure was chosen based on the folding of helices D and E. In the aMD replica chosen, we observed a stabilization of the secondary structure in the lumenal side from 400 ns to the end of the simulation (Supple-mentary Fig. 19). We thus chose a structure at the 400 ns mark as a seed for the following simulation. In order to better compare these new MDs to the previous ones, we used a single initial structure for the following step. The new structure followed the entire minimization/equilibration protocol described above (except for the *in vacuo* minimization), to avoid biases in the comparison with the other trajectories. Two additional 1 µs production simulations (MD7, MD8) were per-formed starting from these conditions.

### Clustering of conformations.
Backbone Ramachandran dihedral angles of the protein, excluding the stromal loop and the first four residues, were used as input features for the clustering. Ramachandran dihedrals ($\phi$ and $\psi$) are the two main degrees of freedom of a peptide bond: $\phi$ is the dihedral angle that allows rotation around the C$_\alpha$–N bond and $\psi$ defines the rotation around the C$_\alpha$–C bond. These two angles define the secondary and tertiary structure of the apoprotein. In order to define an RMSD in the dihedral angle space, we transformed each $\phi$ or $\psi$ angle in its sine and cosine, and then performed a PCA retaining the components that make up 95% of the variance. This procedure is known in the literature as dPCA[45]. A hierarchical agglomerative clustering using the Ward distance measure was performed on the dPCA coordinates, and the number of clusters was determined by visual inspection of the dendrogram. The clustering was performed with the python package *scikit-learn*.

### Quantum chemical calculations of spectra.
For the calculations of the optical spectra, we followed the same protocol as in our previous study of the LHCII complex[46]. Transition energies and intensities of the complex were obtained through an excitonic approach based on the lowest excited state of chlorophylls ($Q_y$) and of carotenoids. Site energies, transition dipoles, and couplings were cal-culated at TDDFT level on configurations of the solvated complex extracted from the MD trajectories. The M06-2X functional was used in combination with the 6-31G(d) basis set. The effects of the protein and the environment were included through a polarizable QM/MM methodology (MMPol)[47,48]: the pigments are treated at TD-DFT level, whereas the rest of the atoms (the protein, membrane, and the solvent) were treated at MM level. A radius of 15 Å was used for the polarization cutoff. The MMPol atoms were described using charge and polariz-ability parameters derived by Wang et al.[49]. In particular, the parameter set based on Thole's linear smeared dipole field tensor was used, in which 1–2 and 1–3 interactions are excluded. A truncated QM model was used for the Chl in which the phytyl chain has been cut and the dangling bond has been saturated with a hydrogen atom. The atoms of the phytyl chain were included as MMPol sites.

The couplings were calculated only between pigments with the intermolecular distance smaller than 30 Å, whereas the other couplings were neglected. The exciton model was coupled to the Redfield for the description of inter-exciton relaxation. The parameters of the spectral densities for the different pigments are reported in the Supplementary Information.

For every replica, 51 structures from the second half of the production run were selected. The structures were selected equidistantly, one every 10 ns. This approach allows us to include a large range of the static disorder present in the system. Before calculating the spectra, we separated fast and slow fluctuations as explained in the Supplementary Methods (see also Supplementary Figs. 20–22 and Supplementary Table 4). We then obtained one spectrum for each structure, which takes into account the slow fluctuations of site energies, couplings, and transition dipoles. This approach is described in detail in the Supplementary Methods. The spectra were finally averaged over all structures belonging to the same cluster.

All the calculations have been performed with a locally modified version of the Gaussian09 package[50] and post-processed with the EXAT program[51].

### Statistics and reproducibility.
As detailed in the Molecular Dynamics section, $n = 6$ independent conventional MD replicas were run from the homology structure. After the aMD equilibration, $n = 2$ additional independent conventional MD replicas were run as explained in the Accelerated MD equilibration section.

Calculations of spectra were based on $n = 51$ uncorrelated structures for each replica. Each cluster spectrum was represented by $n = 51$–153 structures, depending on the number of replicas in each cluster.

The good average reproducibility of the MD simulations after aMD equilibration is demonstrated by the low RMSD of Supplementary Fig. 4.

**Reporting summary**. Further information on research design is available in the Nature Research Reporting Summary linked to this article.

## Data availability
Representative structures of the four clusters are given in Supplementary Data 1. The datasets generated during and/or analyzed during the current study are available as Source Data File in Supplementary Data 2–16.

## Code availability
The custom code used for this study is available from the corresponding authors upon request.

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

## Acknowledgements

The authors acknowledge funding by the European Research Council (ERC), under the grant ERC-AdG-786714 (LIFETimeS).

## Author contributions

Conceptualization: L.C. and B.M.; structural methods: I.G.P. and C.P.; spectral methods: V.S. and C.P.; formal analysis and investigation: I.G.P, V.S., C.P., L.C., and B.M.; writing/figures: I.G.P, V.S., C.P., L.C., and B.M.; funding acquisition: B.M. All authors have read and agreed to the published version of the manuscript.

## Competing interests
The authors declare no competing interests.
