## [Transparent Peer Review File · Communications Biology]

Reviewers' comments:

Reviewer #1 (Remarks to the Author):

The study by Prandi et al. report on the structure of the stress-related LHCSR1 complex in algae and mosses. There is yet no experimentally resolved structure reported in the literature, so this computational prediction can form the basis for future works on LHCSR1. The authors reproduce well the experimental spectroscopic properties of the complex, based on their prediction. This certainly adds to the validity of their models. The original study should be of great interest to those working on photosynthesis, but also to those in the wider field of structural Biology. The methods employed seem appropriate and robust, whereas enough details are given for the work to be reproduced. However I have a couple of points that the authors might want to address. In detail:

(1) In the methods it is reported that "The aMD equilibration was started from the end of MD1, and the coordinates of the complex after 1us were extracted and used as a starting point for two further replicas (MD7-MD8)". In the results the authors claim that "The clustering algorithm has distributed the frames from trajectories MD1-MD6 in three different clusters (Clusters 1-3) while the structures obtained from MD7 and MD8 have been merged in the same cluster (Cluster 4) thus indicating a clear separation of these replicas, which have been pre-equilibrated with aMD, from the others." However, all aMD runs (MD7-MD8) were initiated from the same "MD1" cluster of structures, if not from the exact same final structure of a single trajectory (MD1). Could this bias the result? In my opinion the aMD runs should have been initiated from the structures out of each MD1-MD6 trajectory, or out of the representative clusters 1-3 out of the MD1-6 trajectories, to enhance sampling.

(2) Considering the evolution of the photosynthetic apparatus/ photoprotective mechanisms among different species and given that mosses are placed inbetween algae and higher plants in this trajectory of evolution, the authors should at least briefly discuss their predicted LHCSR1 structure within this context.

Overall I find this contribution interesting, and appropriate for publication in Communications Biology.

Reviewer #2 (Remarks to the Author):

LHCSR1 complex is associated with NPQ processes in algae, and hence is of importance to study. However there are no x-ray or cryo-em structure available. The manuscript attempts to use an integrated computational modeling strategy to predict and validate the structure. With CP29 as a starting point for homology modelling. The structure of the LHCSR1 was then obtained using MD simulation and subsequent clustering. Excitonic levels were calculated and comparisons to experimental linear spectra and CD spectra were made to validate the results. Further studies by pigment deletions or exchange were made for comparison/validation.

The manuscript is mostly clear and detail. The evidence and reasoning are well presented. The results are new and will be interesting for scientist working on photosynthesis. I recommend acceptance after minor revisions below:

The readers may not all be MD scientists or structural scientist, so more specific terms like "Ramachandran dihedral angle" used for clustering may need to be briefly explained.

The comparison to spectra seems to work better for LHCSR (cluster 4) than CP29. For example the both the expt linear and CD spectrum shows a small peak for CP29 at approx 15600cm⁻¹ which does not appear fr the simulated one. Also, the CD experimental peak at 14700 is almost missed by the calculation (on the other hand The CD LHCSR peaks at 14600 matched up for calculated and experimental). Can the authors comment on it?

Typo pg 4 beginning of 2nd paragraph: '\... \mu-long trajectories' should be '\... \mu s-long trajectories'

Reviewer #3 (Remarks to the Author):

In this manuscript, the authors present a modeled structure of a stress-related light-harvesting complex, LHCSR1, which plays an important role in photoprotection in algae and mosses. Using a combination of homology modeling, molecular dynamics, quantum chemistry, and spectral modeling, the authors propose and verify the structure of LHCSR1 of the moss *Physcomitrella patens*. This is particularly interesting because up to now, no experimentally determined structure of LHCSR is available. The modeled structure can provide the basis for a more detailed understanding of the mechanisms of photoprotection in algae and mosses. To facilitate this, it would be beneficial if the coordinates of a representative structure of cluster 4 would be added to the supplementary information.

Starting from the structure of CP29, a homology model is built for LHCSR1 and pigments according to experimental pigment analysis are included in the structure. The structure is relaxed in a model membrane (containing DOPC), the structures are clustered, and pigment orientations are analyzed. Furthermore, the excitonic Hamiltonian for the different clusters is calculated and absorption as well as circular dichroism (CD) spectra are simulated. The overall approach is technically sound but some details of the procedure are missing. The comparison of simulated to experimental spectra supports the conclusions.

Overall, the paper is suitable for *Communications Biology* and of interest to its readership. However, there are several points which should be addressed in a major revision before publication:

(1) Chl a611 and a604 are thought to be present but it is not directly bound to a protein residue. Can the authors identify any stabilizing interactions between Chl a604/a611 and neighboring residues which could explain the assumed high stability of these Chls? Typically, the thylakoid membrane is highly dynamic making the residence of a loosely bound Chl challenging.

(2) On p. 2, the axial ligand of Chl a611 is specified as DPPC lipid, however in the Methods section (Homology modeling, p. 11) it is stated that the coordinates of the DPPG lipid present in CP29 of pdb code 3JCU were used. Please clarify the employed lipid type. If it is DPPG, does the lipid stick to Chl a611 throughout the simulation time?

(3) In ref. 22 of the manuscript, models for the LHCSR1 and LHCSR3 proteins of *Clamydomonas reinhardtii* are presented. How does the model of LHCSR1 presented here compare with the LHCSR1/3 models from ref. 22?

(4) In the presented model, the 81 N-terminal residues are excluded. Are they expected to be intrinsically disordered? What could be their impact on the LHCSR1 dynamics and its regulatory function?

(5) During the aMD simulations, weak restraints were added to the transmembrane helices. Were these restraints released after the aMD simulations for the production runs MD7 and MD8?

(6) From the three aMD replicas, only one final structure was chosen for another 2 unbiased replicas (MD7, MD8). Why was only one starting structure chosen and what was the selection criterion?

(7) Frames from the trajectories MD1-MD6 were clustered in the clusters 1-3. How is the distribution of the replicas to the different clusters? Do all replicas contribute to multiple clusters or does a rare event during the equilibration determine the cluster assignment?

(8) To cluster the structures from the MD trajectories, a principal component analysis (PCA) was performed before the clustering. Was the PCA only used for coordinate transformation or also for dimensionality reduction?

(9) In Fig. S10, the binding site distance of Chl a613 shows sudden jumps in trajectories MD1 and MD3 as well as an unbinding in the long equilibrated structure MD7. Why do some of the jumps not result in unbinding while in the more reliable structure MD7 an unbinding is observed?

(10) To compute the absorption and CD spectra, structures from the 8 replicas were extracted and grouped according to the clustering. How many structures were extracted from each replica? How were these structures selected? How many structures were averaged for the individual clusters after assigning the structures to their respective cluster?

(11) In the caption of Fig. 6, the line types are not ascribed correctly to the depicted spectra or the legends in the plots are wrong. Please clarify.

(12) There is a typo on p. 10: "Hamiltonian a recalculated" should read "Hamiltonian and recalculated".

(13) As mentioned in the Conclusions, the folding and orientation of helices D and E are influenced by the membrane-water interface. The thylakoid membrane contains a large amount of glycolipids which are not present in the model membrane employed here. Could the authors comment on the effect of glycolipids on the helix folding and orientation?

(14) The excitonic Hamiltonian is averaged before calculating the spectra. However, the excitation process is much faster than the structural fluctuations present in the MD trajectories. How large are the fluctuations of the different elements in the Hamiltonian? And to which extent would an averaging of the spectra calculated for each individual excitonic Hamiltonian change the resulting spectra?

(15) Do the authors expect differences in the Chl orientation during the MD simulations when changing Chl a609 to Chl b? If yes, might those differences be able to compensate for the differences in the spectra shown in Fig. 7c/d?

Reviewer #1 (Remarks to the Author):

The study by Prandi et al. report on the structure of the stress-related LHCSR1 complex in algae and mosses. There is yet no experimentally resolved structure reported in the literature, so this computational prediction can form the basis for future works on LHCSR1. The authors reproduce well the experimental spectroscopic properties of the complex, based on their prediction. This certainly adds to the validity of their models. The original study should be of great interest to those working on photosynthesis, but also to those in the wider field of structural Biology. The methods employed seem appropriate and robust, whereas enough details are given for the work to be reproduced. However I have a couple of points that the authors might want to address.

Authors' Reply: We sincerely thank the Reviewer for their positive comments.

In detail:

(1) In the methods it is reported that "The aMD equilibration was started from the end of MD1, and the coordinates of the complex after 1us were extracted and used as a starting point for two further replicas (MD7-MD8)". In the results the authors claim that "The clustering algorithm has distributed the frames from trajectories MD1-MD6 in three different clusters (Clusters 1-3) while the structures obtained from MD7 and MD8 have been merged in the same cluster (Cluster 4) thus indicating a clear separation of these replicas, which have been pre-equilibrated with aMD, from the others." However, all aMD runs (MD7-MD8) were initiated from the same "MD1" cluster of structures, if not from the exact same final structure of a single trajectory (MD1). Could this bias the result? In my opinion the aMD runs should have been initiated from the structures out of each MD1-MD6 trajectory, or out of the representative clusters 1-3 out of the MD1-6 trajectories, to enhance sampling.

Authors' Reply: We thank the Reviewer for raising this point. We have used aMD in order to further refine a structure that was "stuck" in a metastable state towards a more stable state. We expected the aMD to guide the system towards a much more stable structure, which indeed is what happens. We then used the final structure of the aMD as a new starting structure for the entire protocol: the additional MD runs (MD7 and MD8) started from the aMD but then were subjected to the same protocol (heating, equilibration, and production) as the first 6 MDs.

If there were a bias coming from MD1 and propagating through the aMD towards the new unbiased MDs 7 and 8, we would have observed a larger similarity of MD7 and MD8 with MD1 than with all other MDs. However, this is not what we observe. From the dihedral RMSDs of Supplementary Figure 4, MD7 -8 are closer to the last part of MD3 than to the last part of MD1. Moreover, the fact that MD7 and MD8 are closer to each other also suggests that the basin they explore is much more stable, and would have been reached independently of the exact starting point.

(2) Considering the evolution of the photosynthetic apparatus/ photoprotective mechanisms among different species and given that mosses are placed in between algae and higher plants in this trajectory of evolution, the authors should at least briefly discuss their predicted LHCSR1 structure within this context.

Authors' Reply: We thank the Reviewer for this very interesting suggestion. Unfortunately though, it is difficult for us to discuss the predicted LHCSR1 structure within an evolutionary perspective as we do not have access to the LHCSR structure in algae. Instead, in this paper we have tried to clarify similarities and differences with respect to LH complexes in plants. We sincerely hope to progress further in this analysis by revealing the details of LHCSR systems in different organisms.

Overall I find this contribution interesting, and appropriate for publication in Communications Biology.

Reviewer #2 (Remarks to the Author):

LHCSR1 complex is associated with NPQ processes in algae, and hence is of importance to study. However there are no x-ray or cryo-em structure available. The manuscript attempts to use an integrated computational modeling strategy to predict and validate the structure. With CP29 as a starting point for homology modelling. The structure of the LHCSR1 was then obtained using MD simulation and subsequent clustering. Excitonic levels were calculated and comparisons to experimental linear spectra and CD spectra were made to validate the results. Further studies by pigment deletions or exchange were made for comparison/validation.

The manuscript is mostly clear and detail. The evidence and reasoning are well presented. The results are new and will be interesting for scientist working on photosynthesis. I recommend acceptance after minor revisions below.

Authors' Reply: We sincerely thank the Reviewer for their positive comments.

(1) The readers may not all be MD scientists or structural scientist, so more specific terms like "Ramachandran dihedral angle" used for clustering may need to be briefly explained.

Authors' Reply: We thank the Reviewer for this suggestion. With "Ramachandran dihedral angles" we refer to the two relevant dihedral angles, ϕ and ψ , of the protein backbone, which essentially define the secondary and tertiary structure of the apoprotein. To define a RMSD in the dihedral angle space, we transformed each dihedral angle (both ϕ and ψ of each considered residue) in its sine and cosine, and then performed a PCA. In these new coordinates it is possible to both define a RMSD between two conformations and to employ a clustering algorithm.

We changed the relevant part of the Methods section to better clarify:

"Backbone Ramachandran dihedral angles of the protein, excluding the stromal loop and the first 4 residues, were used as input features for the clustering. Ramachandran dihedrals (ϕ and ψ) are the two main degrees of freedom of a peptide bond: ϕ is the dihedral angle that allows rotation around the C α -N bond and ψ defines the rotation around the C α -C atoms of the amino acid residue. These two angles define the secondary and tertiary structure of the apoprotein. In order to define a RMSD in the dihedral angle space, we transformed each ϕ or ψ angle in its sine and cosine, and then performed a PCA retaining the components that make up 95% of the variance. This procedure is known in the literature as dPCA."

(2) The comparison to spectra seems to work better for LHCSR (cluster 4) than CP29. For example, both the expt linear and CD spectrum shows a small peak for CP29 at approx 15600cm⁻¹ which does not appear for the simulated one. Also, the CD experimental peak at 14700 is almost missed by the calculation (on the other hand The CD LHCSR peaks at 14600 matched up for calculated and experimental). Can the authors comment on it?

Authors' Reply: We note that the small negative peak in the CD spectrum and the positive one in the absorption of CP29 at 15600 cm⁻¹ both correspond to a chlorophyll-b signal. It seems that in our simulation the excitation energies of these chlorophylls are underestimated and contribute to the

signal at a lower frequency. Moreover, the simulated CD spectrum of CP29 is somewhat broader than the experimental one, slightly obscuring the double peak character of the CD spectra around 15000 cm^{-1} which is noticeable in the experiment. The broader CD spectra in the low frequency part can be related to the same effect in the absorption spectra, which could be caused by the lineshape which was used for the chlorophylls or by few conformations with low energy exciton states.

We want also to underline that the CD spectra are in general very hard to simulate with QM methods, as the presence of positive and negative signals make the final quality of the spectrum extremely sensitive to even very small inaccuracies in the site energy positions. These simulated spectra can therefore be considered as an excellent match with the experimental spectra considering the precision of the currently used methods.

What is more important is that we do reproduce the specific differences, both in absorption and in CD, when passing from CP29 to LHCSR1. This point lets us believe that our model, by deviating from CP29, goes in the correct direction towards the real LHCSR1 structure.

(2) Typo pg 4 beginning of 2nd paragraph: '... μ -long trajectories' should be '... μ s-long trajectories'

Authors' Reply: We thank the reviewer for noticing the typo. It was corrected in the main text.

Reviewer #3 (Remarks to the Author):

In this manuscript, the authors present a modeled structure of a stress-related light-harvesting complex, LHCSR1, which plays an important role in photoprotection in algae and mosses. Using a combination of homology modeling, molecular dynamics, quantum chemistry, and spectral modeling, the authors propose and verify the structure of LHCSR1 of the moss *Physcomitrella patens*. This is particularly interesting because up to now, no experimentally determined structure of LHCSR is available. The modeled structure can provide the basis for a more detailed understanding of the mechanisms of photoprotection in algae and mosses. To facilitate this, it would be beneficial if the coordinates of a representative structure of cluster 4 would be added to the supplementary information.

Starting from the structure of CP29, a homology model is built for LHCSR1 and pigments according to experimental pigment analysis are included in the structure. The structure is relaxed in a model membrane (containing DOPC), the structures are clustered, and pigment orientations are analyzed. Furthermore, the excitonic Hamiltonian for the different clusters is calculated and absorption as well as circular dichroism (CD) spectra are simulated. The overall approach is technically sound but some details of the procedure are missing. The comparison of simulated to experimental spectra supports the conclusions.

Overall, the paper is suitable for *Communications Biology* and of interest to its readership. However, there are several points which should be addressed in a major revision before publication:

Authors' Reply: We thank the Reviewer for the constructive and positive comments.

We have included the coordinates of 5 representative structures of cluster 4 in a supplementary file together with the other clusters. We have also changed the main text according to the following points, and we hope that we have substantially improved the manuscript so that it merits publication.

(1) Chl a611 and a604 are thought to be present but it is not directly bound to a protein residue. Can the authors identify any stabilizing interactions between Chl a604/a611 and neighboring residues which could explain the assumed high stability of these Chls? Typically, the thylakoid membrane is highly dynamic making the residence of a loosely bound Chl challenging.

Authors' Reply: We thank the Reviewer for this suggestion. Chlorophyll a604 is bound to the backbone oxygen of Gly109 through a water molecule, and it forms an additional hydrogen bond with the backbone of the loop His136, as shown in Supplementary Figure 17. The same binding through a water molecule is present also in LHCII, CP29, and other complexes in the LHC family. Importantly, this binding is stable throughout the whole MD simulation in LHCSR (Supplementary Figure 11) and for all the other systems (CP29 and LHCII).

We added a comment in the relevant paragraph of the Results section:

“Analogously to CP29 and LHCII, a604 is bound to the backbone oxygen of Gly109 through a water molecule.”

The binding of chlorophyll a611 occurs through the axial ligand lipid (DPPG) which is then bound either to a Lys (in CP29 and LHCII) or to an Asn (LHCSR1). The binding between a611 and lipid is very

stable and is present during the whole MD simulation. We added in Supplementary Figure 11 a panel showing the binding distance between a611 and the DPPG along the MD replicas.

We also added the following comment in the Results section: “Finally, the axial binding between a611 and the DPPG lipid remained stable and tight in all simulations except MD1 and MD3 (Supplementary Figure 11).”

(2) On p. 2, the axial ligand of Chl a611 is specified as DPPC lipid, however, in the Methods section (Homology modeling, p. 11) it is stated that the coordinates of the DPPG lipid present in CP29 of pdb code 3JCU were used. Please clarify the employed lipid type. If it is DPPG, does the lipid stick to Chl a611 throughout the simulation time?

Authors' Reply: The reviewer is right: the employed lipid is DPPG, taken from the CP29 structure as we described. DPPC is a typo and we changed it in the manuscript. We confirm that the Chl a611 is stabilized by the presence of the lipid and their interactions are mainly maintained throughout the simulations. As explained above, we included a new panel in Supplementary Figure 11 showing that the binding is stable.

We also added the following comment in the Results section: “Finally, the axial binding between a611 and the DPPG lipid remained stable and tight in all simulations except MD1 and MD3 (Supplementary Figure 11).”

(3) In ref. 22 of the manuscript, models for the LHCSR1 and LHCSR3 proteins of *Chlamydomonas reinhardtii* are presented. How does the model of LHCSR1 presented here compare with the LHCSR1/3 models from ref. 22?

Authors' Reply: Unfortunately, we do not have enough information on the *C. reinhardtii* LHCSR1/3 models from Ref. 22 to properly compare our models. However, by looking at the Figures of Ref. 22, we notice that both LHCSR models are extremely similar to the template (which in that case is Lhcb1). Similarly, our initial homology model is very similar to the parent CP29. For *P. patens* LHCSR1, our results suggest a substantial rearrangement of the structure, especially in the luminal side. By analogy with our results, also for *C. reinhardtii* LHCSR1/3 we expect that the real structure will deviate substantially from the models of Ref. 22.

We added the following to the discussion:

“Perozeni et al. have presented homology models of LHCSR1 and LHCSR3 from *Chlamydomonas Reinhardtii* based on Lhcb1 as a template. As in our case, their homology models follow very closely the template structure. Our MD results however suggest caution in using these models, as the real structure of LHCSR may deviate substantially from the currently available template structures.”

(4) In the presented model, the 81 N-terminal residues are excluded. Are they expected to be intrinsically disordered? What could be their impact on the LHCSR1 dynamics and its regulatory function?

Authors' Reply: The N-terminal residues of LHCSR1 were excluded in this work due to the very low identity in that region. This region is quite disordered in the Lhcb proteins such as CP29 and LHCII, therefore we expect a similar pattern also for LHCSR1.

In evaluating the impact of this part on the LHCSR1 dynamics, we should first notice that all the chlorophylls are bound to the main transmembrane helices, which are very stable. We do not expect the N-terminus to change the mutual orientation and distances between the pigments. If these residues interact with the pigments, they might lead to slightly larger site energy fluctuation for the nearby chlorophyll a609. Based on these considerations, we believe that the main optical properties of the system will not be affected by the N terminal residues.

The regulatory function of LHCSR1 is driven by protonation of the luminal side of the protein. As the N-terminus is in the stromal side, we expect its possible impact on the NPQ regulation to be marginal.

(5) During the aMD simulations, weak restraints were added to the transmembrane helices. Were these restraints released after the aMD simulations for the production runs MD7 and MD8?

Authors' Reply: Yes, the additional restraints were used only for the aMD to avoid unfolding of the transmembrane helices, which are already well equilibrated in the first 6 replicas (MD1-MD6). In practice, we used these restraints in order to avoid changes in the already high-confidence regions of the model.

For the subsequent conventional MD simulations (MD7-8), we followed the initial protocol from the beginning, using the protein structure from the aMD as a new starting structure, and repeated the heating and equilibration phases. In the production phase, MD7 and MD8 were ran exactly in the same way as MD1-6, and in particular without any restraint.

(6) From the three aMD replicas, only one final structure was chosen for another 2 unbiased replicas (MD7, MD8). Why was only one starting structure chosen and what was the selection criterion?

Authors' Reply: One of the aMD was chosen because it showed a stable secondary structure from 400 ns to 1000 ns of simulation time for both helices E and D (See the new Supplementary Figure 21). We thus chose the first frame of this newly stabilized structure, around the 400 ns mark.

As a compromise between sampling accuracy and computational time, we decided to start two MDs from the same (aMD-equilibrated) initial structure (MD7 and MD8). In this way, MD7-8 can be compared more realistically, because they start from the same initial structure (as MD1-6) and they undergo the same protocol as the previous MDs.

We added the following clarification to the Methods section: "In the aMD replica chosen, we observed a stabilization of the secondary structure in the luminal side from 400 ns to the end of the simulation (Supplementary Figure 21). We thus chose a structure at the 400 ns mark as a seed for the following

simulation. In order to better compare these new MDs to the previous ones, we used a single initial structure for the following step.”

(7) Frames from the trajectories MD1-MD6 were clustered in the clusters 1-3. How is the distribution of the replicas to the different clusters? Do all replicas contribute to multiple clusters or does a rare event during the equilibration determine the cluster assignment?

Authors' Reply: We thank the Reviewer for this question. Each replica (1-6) contributes only to a single cluster. This can be seen already at the level of dihedral RMSD (Supplementary Figure 4) where frames coming from the same replica are closer than frames from different replicas (at least for MD1-MD6).

How replicas cluster together can be explained by the fact that the luminal helices in the initial structure are far from the equilibrium conformation, therefore a small fluctuation during the equilibration propagates to large differences between the replicas (clusters). We additionally note that the clustering was performed on the last 500 ns of each replica, therefore the first half of the “production” can be thought as an extended equilibration. This can be also seen in Supplementary Figure 4, where the “well equilibrated” replicas MD7-8 are much more similar than the initial replicas even within the same cluster because the MD7-8 replicas start from the structure close to the equilibrium one (the same potential well).

We added this remark to the Results section: “Notably, each replica MD1-MD6 contributes to only one of clusters 1-3, meaning that our conventional MD replicas tend to diverge from the initial state and get stuck in different metastable states.”

(8) To cluster the structures from the MD trajectories, a principal component analysis (PCA) was performed before the clustering. Was the PCA only used for coordinate transformation or also for dimensionality reduction?

Authors' Reply: The PCA analysis was used for both coordinate transformation and for the dimensionality reduction, in order to reduce the noise for the clustering. In practice, we included all the first N principal components that explain 95% of the variance, which means that we only compressed the initial dataset with minimal loss. We have checked that, by including all principal components, the clustering results are identical, and the dihedral RMSD matrix (Supplementary Figure 4) does not significantly change.

In order to clarify this and other details, we changed the relevant part of the Methods section: “Backbone Ramachandran dihedral angles of the protein, excluding the stromal loop and the first 4 residues, were used as input features for the clustering. Ramachandran dihedrals (φ and ψ) are the two main degrees of freedom of a peptide bond: φ is the dihedral angle that allows rotation around the C α -N bond and ψ defines the rotation around the C α -C atoms of the amino acid residue. These two angles define the secondary and tertiary structure of the apoprotein. In order to define a RMSD in the dihedral angle space, we transformed each φ or ψ angle in its sine and cosine, and then

performed a PCA retaining the components that make up 95% of the variance. This procedure is known in the literature as dPCA.”

(9) In Fig. S10, the binding site distance of Chl a613 shows sudden jumps in trajectories MD1 and MD3 as well as an unbinding in the long equilibrated structure MD7. Why do some of the jumps not result in unbinding while in the more reliable structure MD7 an unbinding is observed?

Authors' Reply: The increase in distance between chlorophyll a613 and the binding Gln 226 site arises from the presence of a water molecule between the Mg of the chlorophyll and the Gln. In this conformation, the chlorophyll a613 is still bound, not directly to the Gln but through the water molecule (See Supplementary Figure 22). This conformation is further stabilized by a hydrogen bond between atom O1D of the chlorophyll and Ser 230.

We added in Supplementary Figure 22 a visualization of this indirect binding and we also added the following comment to the main text:

“We note that in MD7 there are some differences in the distance between Chl a613 and its binding site, the Gln226 residue (Supplementary Figure 10), which is due to the presence of a water molecule between the residue and the chlorophyll, causing chlorophyll a613 to be indirectly bound to Gln in some time windows. This conformation is further stabilized by a hydrogen bond between the O1D atom of the chlorophyll and Ser 230.”

(10) To compute the absorption and CD spectra, structures from the 8 replicas were extracted and grouped according to the clustering. How many structures were extracted from each replica? How were these structures selected? How many structures were averaged for the individual clusters after assigning the structures to their respective cluster?

Authors' Reply: For every replica, 51 structures from the second half of the production run were selected. The structures were selected equidistantly, each every 10ns. This approach allows us to include a large range of the static disorder present in the system. All the structures belonging to a cluster were taken and used to compute the cluster average.

We added the following explanation to the Methods section:

“For every replica, 51 structures from the second half of the production run were selected. The structures were selected equidistantly, one every 10ns. This approach allows us to include a large range of the static disorder present in the system. Before calculating the spectra, we separated fast and slow fluctuations as explained in the Supplementary Methods. We then obtained one spectrum for each structure, which takes into account the slow fluctuations of site energies, couplings, and transition dipoles. The spectra were finally averaged over all structures belonging to the same cluster.”

(11) In the caption of Fig. 6, the line types are not ascribed correctly to the depicted spectra or the legends in the plots are wrong. Please clarify.

Authors' Reply: We agree with the reviewer, that in the figure caption the line assignment was wrong. The Figure 6 caption was corrected to: “Spectra computed for the LHCSR1 cluster 4 and CP29 (solid

lines), compared with the experiment (dashed lines). a) absorption spectra, b) CD spectra. Note that calculated Chl site energies were all shifted by the same amount (-1381 cm^{-1}) to account for the error of the quantum chemical method; this shift was determined independently on LHCII (see Methods). The experimental spectrum for LHCSR1 was taken from Ref 20 and the CP29 one was taken from Ref 29.”

(12) There is a typo on p. 10: "Hamiltonian a recalculated" should read "Hamiltonian and recalculated".

Authors' Reply: We thank the reviewer for noticing the typo. It was corrected in the main text.

(13) As mentioned in the Conclusions, the folding and orientation of helices D and E are influenced by the membrane-water interface. The thylakoid membrane contains a large amount of glycolipids which are not present in the model membrane employed here. Could the authors comment on the effect of glycolipids on the helix folding and orientation?

Authors' Reply: We agree that the type of lipids used here could have different effects on some properties of the protein. However, the helix folding and orientation are driven by the hydrophobic/hydrophilic sides of the system, and as such depend only on the fact that there is a hydrophobic core in the membrane. Therefore, these structural features should not depend on the composition of the membrane and on the type of lipid.

(14) The excitonic Hamiltonian is averaged before calculating the spectra. However, the excitation process is much faster than the structural fluctuations present in the MD trajectories. How large are the fluctuations of the different elements in the Hamiltonian? And to which extent would an averaging of the spectra calculated for each individual excitonic Hamiltonian change the resulting spectra?

Authors' Reply: We apologize for the misunderstanding: we have not averaged the Hamiltonian over all frames for the calculation of the spectra, but only for obtaining the exciton levels in Figure 5. For the simulation of the optical spectra, only the “geometrical” contribution to the site energies was averaged, whereas the rest of the quantities in the Hamiltonian, such as couplings and the environmental contribution to the excitation energy, were used as obtained for each configuration of the system. The fluctuations of the environmental contribution to the excited energy were assigned to the static disorder. All details on how we treated the site energy fluctuations are explained in the Supplementary Methods, section “Site energies and static disorder”. The optical spectra were then computed for each configuration separately and the final spectra were obtained from averaging of the spectra over the configurations belonging to the same cluster. In this way we simulate the experimental conditions, where the measurements are performed on an ensemble of the structures. The approach is described in more detail in our previous work on LHCII referenced in the main text (ref. 47) - “For the calculations of the optical spectra, we followed the same protocol as in our previous study of the LHCII complex 47”.

We also changed the relevant bit in the Methods section to better explain our procedure:

“For every replica, 51 structures from the second half of the production run were selected. The structures were selected equidistantly, one every 10ns. This approach allows us to include a large range of the static disorder present in the system. Before calculating the spectra, we separated fast and slow fluctuations as explained in the Supplementary Methods. We then obtained one spectrum for each structure, which takes into account the slow fluctuations of site energies, couplings, and transition dipoles. The spectra were finally averaged over all structures belonging to the same cluster.”

(15) Do the authors expect differences in the Chl orientation during the MD simulations when changing Chl a609 to Chl b? If yes, might those differences be able to compensate for the differences in the spectra shown in Fig. 7c/d?

Authors' Reply: We thank the Reviewer for this suggestion. We have not found any residue which may form hydrogen bonds with the carbonyl group of Chl b and therefore suggest a different orientation. In LHCII, the carbonyl group of Chl b609 is hydrogen bonded to the Gln of the same helix. However, the orientation LHCII Chl b609 is the same as the orientation of Chl a609 in CP29. Therefore, there is probably an additional restriction on its orientation imposed by the geometry of the surrounding residues and chlorophylls, whereas the possibility of an H-bond to the carbonyl group may increase the selectivity towards Chl b. Based on this analysis we do not expect a different orientation of Chl b if it was present in the LHCSR.

In addition, we note that the site energy difference between Chl a and Chl b would completely break the exciton delocalization, even if a more favorable orientation (and thus a larger coupling) was possible between a603 and a609.

We added the following comment to the Results section:

“The effect on the spectrum is due to a reduction of the exciton delocalization between Chls 603 and 609 when Chl a609 is substituted by a Chl b, which has a significantly higher excitation energy. Our calculation is based on the assumption that Chl b609 would have the same orientation as the corresponding Chl a609. However, the main effect of reducing exciton delocalization does not depend on the exact position or orientation of Chl b.”

REVIEWERS' COMMENTS:

Reviewer #1 (Remarks to the Author):

The authors have adequately addressed the points raised by the reviewers and the majority of their answers are reflected in the amended manuscript. Therefore, I recommend publication.

Reviewer #3 (Remarks to the Author):

I thank the authors for carefully addressing all my issues and for adding more details of the modeling protocol in the revised manuscript. Therefore, I recommend publication as is.